# Sleeve Gastrectomy-Induced Body Mass Index Reduction Increases the Intensity of Taste Perception’s and Reduces Bitter-Induced Pleasantness in Severe Obesity

**DOI:** 10.3390/jcm11143957

**Published:** 2022-07-07

**Authors:** Sara Rurgo, Elena Cantone, Marcella Pesce, Eleonora Efficie, Mario Musella, Barbara Polese, Barbara De Conno, Marta Pagliaro, Luisa Seguella, Bruna Guida, Giuseppe Esposito, Giovanni Sarnelli

**Affiliations:** 1Department of Clinical Medicine and Surgery, University of Naples “Federico II”, Via Pansini 5, 80131 Naples, Italy; sara.rurgo@unina.it (S.R.); marcella.pesce@unina.it (M.P.); elefficie@hotmail.com (E.E.); barbara.polese@gmail.com (B.P.); barbara.deconno@gmail.com (B.D.C.); martapagliaro@iol.it (M.P.); bruna.guida@unina.it (B.G.); 2Department of Neuroscience, Reproductive and Odontostomatologic Science, ENT Section, ‘Federico II’ University of Naples, 80131 Naples, Italy; elena.cantone@unina.it; 3Advanced Biomedical Sciences Department, Naples “Federico II” University, AOU “Federico II”—Via S. Pansini 5, 80131 Naples, Italy; mario.musella@unina.it; 4Department of Physiology and Pharmacology, Faculty of Pharmacy and Medicine, Sapienza University of Rome, 00161 Rome, Italy; luisa.seguella@uniroma1.it (L.S.); giuseppe.esposito@uniroma1.it (G.E.); 5UNESCO Chair on Health Education and Sustainable Development, “Federico II” University, 80131 Naples, Italy

**Keywords:** obesity, bariatric surgery, taste perception, weight loss

## Abstract

*Background*: The sense of taste is involved in food behavior and may drive food choices, likely contributing to obesity. Differences in taste preferences have been reported in normal-weight as compared to obese subjects. Changes in taste perception with an increased sweet-induced sensitivity have been reported in surgically treated obese patients, but data regarding the perception of basic tastes yielded conflicting results. We aimed to evaluate basic taste identification, induced perception, and pleasantness in normal-weight controls and obese subjects before and after bariatric surgery. *Methods*: Severe obese and matched normal weight subjects underwent a standardized spit test to evaluate sweet, bitter, salty, umami, and sour taste identification, induced perception, and pleasantness. A subset of obese subjects were also studied before and 12 months after sleeve gastrectomy. *Results*: No significant differences in basic taste-induced perceptions were observed, although a higher number of controls correctly identified umami than did obese subjects. Sleeve-gastrectomy-induced weight loss did not affect the overall ability to correctly identify basic tastes but was associated with a significant increase in taste intensities, with higher scores for sour and bitter, and a significantly reduced bitter-induced pleasantness. *Conclusions*: The perception of basic tastes is similar in normal-weight and severely obese subjects. Sleeve-gastrectomy-induced weight loss significantly increases basic taste-induced intensity, and selectively reduces bitter-related pleasantness without affecting the ability to identify the tastes. Our findings reveal that taste perception is influenced by body mass index changes, likely supporting the hypothesis that centrally mediated mechanisms modulate taste perception in severe obesity.

## 1. Introduction

Obesity is a chronic and progressive disease associated with significant morbidity and mortality that has become a worldwide epidemic condition. Globally, a total of 1.9 billion and 609 million adults were estimated to be overweight and obese in 2015, respectively, representing approximately 39% of the world’s population [1]. This increasing prevalence has serious implications for the risk of metabolic disorders [2]. Obese patients (BMI ≥ 30 kg/m^2^), especially morbid ones (BMI is ≥40 kg/m^2^ or ≥35 kg/m^2^ plus one or more co-morbid conditions), require extensive health care due to the association with related medical conditions such as hypertension, diabetes mellitus, hypercholesterolemia, fatty liver disease, obstructive sleep apnea, and kidney and cardiovascular diseases [1,2,3]. Since increased food intake and obesity are related to food preferences, especially for high-calorie foods, gustatory sensitivity seems to determine food choices, amount consumed, and urge to eat [4]. Therefore, gustatory sensitivity directly affects the nutritional status of an individual [4,5,6]. For these reasons, the sense of taste, together with neuronal and hormonal factors, is one of the principal factors determining food intake and the emergence of obesity [1,5,6]. Moreover, psychological aspects affecting hypothalamic signaling, gut hormones, bile acids, and gut microbiota coalesce to exert a profound influence on eating behavior [7]. For instance, taste guides organisms to identify and consume nutrients while avoiding toxins and indigestible materials. For humans, this means recognizing and distinguishing sweet, umami, sour, salty, and bitter—the so-called “basic” tastes [8,9,10]. Taste receptors (TRs) are found in oral and extra-oral sites, as well as in the digestive system, where they are localized in the epithelial cells, including the enteroendocrine (EEC) cells of the gastrointestinal (GI) mucosa lining, and play a critical role in integrating inputs from luminal content and regulating food intake through a gut–brain pathway [11]. Although taste is one of the most important factors influencing feeding behaviors, the number of studies on the gustatory sensitivity of obese patients is still scarce and heterogeneous [12]. Most of the studies have focused on the impaired taste sensitivity in obese subjects and on the change in sweet taste perception [12,13,14,15,16]. The most effective long-term treatment for obesity and its associated comorbidities is bariatric surgery [17,18]. Recent data demonstrated that in surgically treated obese patients, there is a change in taste sensitivity, with an increased sweet taste sensitivity and a decreased preference for sweet tastants, respectively [12,15]. However, only a few studies investigated the effect of obesity and bariatric surgery on the sensitivity to other basic tastes, and data regarding taste recognition thresholds yielded conflicting results [12]. In addition, studies that evaluate basic taste sensitivity were carried out a few months after bariatric surgery, mostly in heterogeneous populations of patients [19,20]. In the present study, we aimed to compare basic taste perceptions in severely obese subjects with an age- and sex-matched population of subjects with normal weight and to evaluate the long-term effects of sleeve gastrectomy-induced weight loss on basic taste identification, induced perception, and pleasantness in a well-selected population of obese individuals.

## 2. Materials and Methods

### 2.1. Subjects

Forty healthy volunteers with normal BMI (14 males and 26 females, mean age 29.7 ± 11; BMI 23.2 ± 3) and seventy-four severely obese subjects (30 males and 44 females, mean age 33.4 ± 9 years, BMI 46 ± 7.4 kg/m^2^) were enrolled. All obese subjects underwent a full clinical examination including ears, nose, and throat (ENT), biochemical, and instrumental evaluation before being considered an eligible candidate for a sleeve gastrectomy (SG). Subjects suffering from diabetic neuropathy, smokers, as well as patients using drugs with central neurological effects were excluded from the study. All subjects were informed about the purpose, nature, and risks of the study before giving their written consent, and the study was approved by the ethical committee of the University of Naples “Federico II”.

### 2.2. Basic Tastes Testing 

All subjects underwent a standardized test to evaluate their ability to identify sweet, bitter, salt, umami, and sour tastes, and to score their relative intensity and pleasantness [10]. In brief, 5 mL of the following substances were randomly administered and served as specific taste agonists: solutions of 10 mM of quinine hydrochloride (bitter), 30 mM of acesulfamate K (sweet), 120 mM sodium chloride (salt), 30 mM of mono potassium glutate plus 0.5 mM of inosine 5′-mono phosphate (umami), and 50 mM of citric acid (sour). After an overnight fast, at 8:30 in the morning, all subjects underwent a taste test to evaluate their ability to correctly identify the basic tastes, to score the intensity, and to indicate the pleasantness of their perception. Taste testing was performed after an overnight fast, at 8:30 in the morning, and took place in a silent room at a warm temperature (20 °C). All solutions used in the study were colorless and retained in similar plastic 5 mL tubes with a protective cap and numbered with a key that was decoded only at the end of each session. The solutions were stored in a refrigerator at a cool temperature of 4 °C and were presented to participants less than 5 min before the test. A maximum of two subjects performed the evaluation at the same time in order to keep a silent atmosphere. Each subject was instructed in the “sip-and-spit” technique; subjects first rinsed their mouth with fresh water, drank a sip of the first solution, turned it in their mouth, assessed the solution, then spit it out, and they used the same procedure for the next solution. After each stimulus, the subjects were invited to rinse their mouth with water to recover a neutral taste sensation on their tongue^14^. In brief, for each solution, subjects first had to identify the taste quality from a list of five descriptors, i.e., sweet, sour, salty, bitter, and umami, and then placed a mark on a visual-analogue scale (VAS) in order to indicate the perceived intensity rating and pleasantness, respectively (0 = not perceived, 100 mm = maximum perceived). 

In 32 subjects of the original cohort of obese subjects, the test was performed at baseline and one year after the sleeve gastrectomy; of the original cohort of 74 obese subjects, 31 were lost at follow-up, 5 denied the informed consent, and 6 subjects refused surgical treatment.

### 2.3. Data Analysis 

Taste-induced intensity of perception and pleasantness were recorded on a 100 mm visual analog scale (VAS) and expressed as mean ± SD. The ability of subjects to correctly identify the different basic tastes was indicated as a percentage and analyzed by Fisher-exact test. Unpaired or paired Student’s *t*-test was used to evaluate the difference in taste-induced intensity and pleasantness between controls and obese patients and in the subset of obese patients before and after surgery, respectively. A *p* value < 0.05 was considered statistically significant. 

## 3. Results

### 3.1. Basic Tastes Identification, Relative Intensity and Pleasantness-Induced Perception in Control and Obese Subjects

Demographic data of the two populations are summarized in Table 1. No significant differences were observed in terms of age and gender prevalence, whereas obese subjects had a significantly higher BMI than controls.

Cumulative analysis revealed that control and obese subjects were similarly able to correctly identify the administered tastants (82 vs. 82.4%, respectively), while the analysis of single tastes revealed that a higher percentage of controls correctly identify umami as compared to obese (83 vs. 61%, *p* < 0.05). No significant differences were observed for the other tastes (Figure 1). 

Taste-induced intensity was similar in the two groups, with bitter and sweet being the most intensely perceived tastes, while umami was the least perceived in both controls and obese subjects. The intensity scores for each of the basic tastes are summarized in Table 2. 

Similarly, no significant differences were observed in the taste-induced pleasantness between the two groups of subjects (Figure 2). 

### 3.2. Weight-Loss Induced Basic Tastes Identification, Relative Intensity and Pleasantness-Induced Perception in Obese Subjects

In all participants, sleeve gastrectomy induced a significant BMI reduction after one year (46 ± 6 vs. 33 ± 2, *p* < 0.001). The overall ability to correctly identify basic tastes was unchanged by weight loss (78% vs. 73%, respectively), with salty and umami being the less correctly identified tastes after sleeve gastrectomy. The percentage of patients correctly identifying the different tastes was not significantly changed after the surgery and is reported in Figure 3. 

Conversely, surgery-induced weight loss was associated with a significant increase in overall taste intensities (70 ± 19 vs. 79 ± 15 mm, *p* < 0.01), with higher scores for sour and bitter (72 ± 32 vs. 91 ± 20 and 83 ± 31 vs. 93 ± 2 mm, *p* < 0.01 and *p* < 0.05, respectively); no significant differences were observed for the other tastes (Figure 4). 

Scores for sour-, salty-, umami-, and sweet-induced pleasantness were similarly unchanged, while bitter-induced pleasantness was significantly and dramatically reduced by weight loss (Figure 5).

## 4. Discussion

Taste is a complex sensory modality involved in feeding behavior and plays an important role in the development of dietary habits and the appeal for high-calorie, palatable foods rich in fats and/or sugars [4]. Evidence suggests that taste preferences differ as a function of body mass index, and the enjoyment of food may vary with body weight changes [4,5]. Generally, sweet taste is associated with energy-dense foods, whereas bitter taste is aversive and related to toxic compounds, suggesting that obese individuals and normal-weight subjects differ in taste preferences [7,21]. However, the correlation between body weight and basic taste qualities has been debated for bitter and sweet perception but largely neglected other basic tastes [12]. Some authors have reported that increases in BMI are associated with a decreased taste sensitivity, which may affect the ability to identify the different basic tastes taste in obesity [22]. Taste may contribute to the development of obesity; however, it is difficult to assess whether altered taste sensitivity causes hyperphagia and weight gain or vice versa [4]. However, this reduced taste sensitivity was found to be reversible with weight loss due to surgery [12]. In fact, studies of bariatric surgery in humans have suggested that taste alterations and food preferences are reversible and, consequently, may represent secondary effects of obesity [5]. By using a standardized test exploring the “taste function”, here, we report that the overall ability to identify the different basic tastes and the related intensity of induced perception and pleasantness did not differ between a large and well-selected cohort of severely obese and age- and sex-matched normal-weight subjects. Our data demonstrated that at least 80% of subjects were able to correctly identify the administered tastants. In particular, subjects were able to correctly identify mostly bitter and sour substances, and this is in line with the observation that these two taste modalities are commonly associated with an aversive valence, alerting individuals against potentially harmful substances [8]. However, only 61% of obese subjects were able to correctly identify the umami tastant. Previous studies have suggested that umami contributes to satiety and may likely influence calorie consumption, especially in individuals with obesity [23]. It is also reported that obese subjects presented reduced sensitivity and a higher preference for umami compared to individuals without obesity [24,25]. Since we did not find any significant differences in both umami-induced sensitivity and pleasantness between obese and controls, it is reasonable to conclude that this apparent discrepancy is due to sociocultural and racial differences. In fact, although umami is particularly appreciated in Japanese cuisine, it is not familiar in the European population, where there is poor awareness of the umami taste [26]. 

Currently, the most effective treatment for obesity is bariatric surgery [17,18]. According to the literature, alterations in food preferences and choices, in particular, the shift away from calorie-dense foods, may contribute to the long-term benefits of surgery, likely reflecting changes in gustatory function [15,16,27,28,29]. An overall increased sensitivity to taste stimuli has been reported in surgically treated obese patients, but very little is known about the effect of weight changes on the perception of all five basic tastes [12]. An increased sensitivity to sweetness and a reduced appetitive reward value for sweet stimuli have been reported to change shortly after bariatric surgery procedures [12]. However, the studied population and methods are heterogeneous, with results being affected by the time after surgery and the different surgical procedures. In order to provide homogenous data about weight-loss-induced changes in taste function, here, we evaluated taste identification ability, perceived intensity, and pleasantness related to the five basic tastes in a large population of severely obese subjects before and one year after sleeve gastrectomy. We observed a significant postoperative reduction in BMI following sleeve gastrectomy but no differences in the ability to identify each of the basic tastes. This finding demonstrated that there were no differences in taste identification with body weight changes and confirmed what was recently reported by Berro et al., who similarly reported no changes six months after sleeve gastrectomy [30].

As already reported by other authors [12], an overall increased taste sensitivity was observed in our obese subjects who lost weight. This likely reflects the improvement of tongue tastebud function by the reduction in obesity-associated inflammation [7,31,32]. Notably, we found a significant and specific increase in the sour- and bitter-induced intensities of perception associated with weight loss. This evidence is in accordance with the recent literature reporting a higher sensitivity to sour and bitter tastes in patients with lower body weight [33]. Although less explored in the context of obesity, sour taste may play a role in food selection and consumption, and our data seem to confirm these observations [4,5,33]. Our findings were also in line with what was reported in the literature, in which bitter taste seems to be involved in feeding behavior and obesity. For instance, some investigators reported that the capacity to detect fatty acids was directly associated with bitter responsiveness, and obese individuals were less sensitive to or had no sensory detection of dietary fat and, therefore, might eat more lipids [34]. Furthermore, the subjects who were unable to taste bitterness due to genetic variations of the bitter taste receptor TAS2R38 were supposed to consume more dietary fat and, therefore, might develop obesity [34]. Other studies reported that bitter taste might affect various brain regions implicated in appetite reduction, and functional magnetic resonance imaging (fMRI) studies have shown that bitter taste was associated with stronger recruitment of different cortical areas, whose responses to bitterness may elicit appetite reduction [35]. Likewise, electroencephalographic (EEG) studies have further demonstrated that bitterness reduces appetitive ratings for high-caloric food images. This evidence supports our findings and, in particular, the observed higher sensitivity to bitter taste scored by our obese patients after weight loss. So, the change in sour and bitter taste perception after bariatric surgery might clarify the underlying mechanism for weight loss. Taste-induced pleasantness has a dramatic impact on food choice and preference. In our hands, obese patients showed a long-term and significant reduction in bitter-related pleasantness. Our data supported previous literature suggesting that in obese patients, following bariatric surgery, there is an alteration of hedonistic responses [36]. Previous fMRI studies showed an increased activation of sensory brain areas as well as areas responsible for reward in response to taste stimuli [37]. We could suppose that after weight loss, patients rated the bitter less pleasant than they perceived before surgery because they perceived it as more intense. Conversely, according to another interpretation, we could speculate that changes in eating behavior are not directly due to taste acuity but to a different reward value attributed to taste stimuli [38,39]. Although our study has some limitations because we did not correlate the changes in taste sensitivity with the dietary habits of patients nor with hormonal variations, here, we provide evidence showing that while the perception of basic tastes is similar in normal-weight and severely obese subjects, sleeve-gastrectomy-induced weight loss induces long-lasting changes in taste-induced intensity and selectively reduces bitter-related pleasantness without affecting the ability to identify the tastes.

## 5. Conclusions

Taste function is dramatically affected by weight loss and may be one of the underlying mechanisms for changing food preferences, further contributing to weight loss and its maintenance in obese subjects. The improved knowledge about peripherally and centrally mediated mechanisms modulating taste perception will pave the way for addressing a gustatory function in individualized obesity treatment approaches.

## Figures and Tables

**Figure 1 jcm-11-03957-f001:**
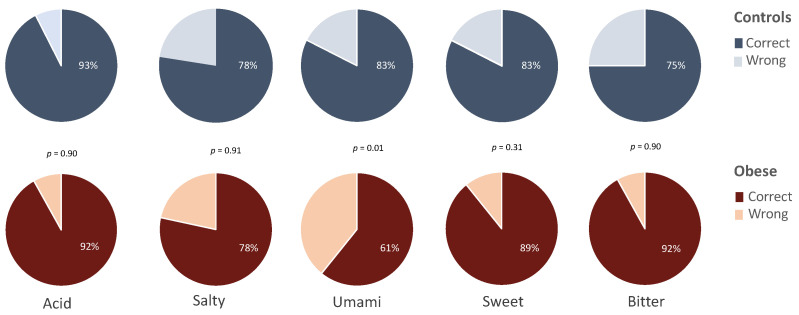
Percentage of controls (*n* = 40) and obese subjects (*n* = 74) correctly identifying the basic tastes.

**Figure 2 jcm-11-03957-f002:**
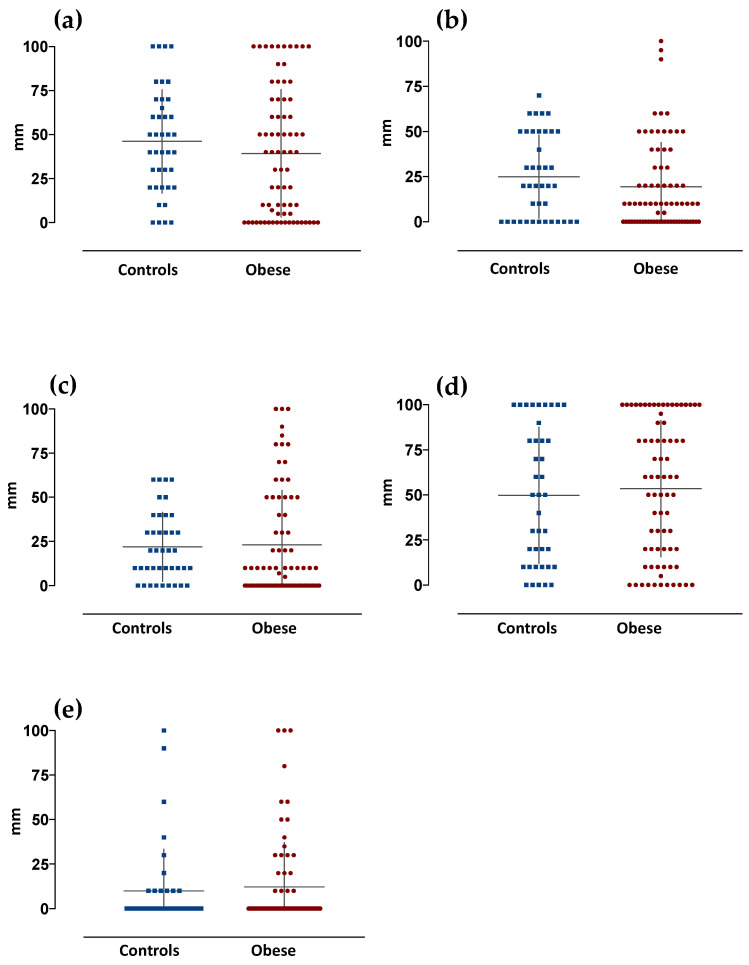
Basic taste-induced pleasantness in control (*n* = 40) and obese subjects (*n* = 74). Graphs are relative to values of taste-induced pleasantness scored by subjects. Data are expressed as mm as measured on a 100 mm VAS scale (mean ± SD), and relative to the different basic tastes (**a**) acid, (**b**) salty, (**c**) umami, (**d**) sweet and (**e**) bitter, respectively.

**Figure 3 jcm-11-03957-f003:**
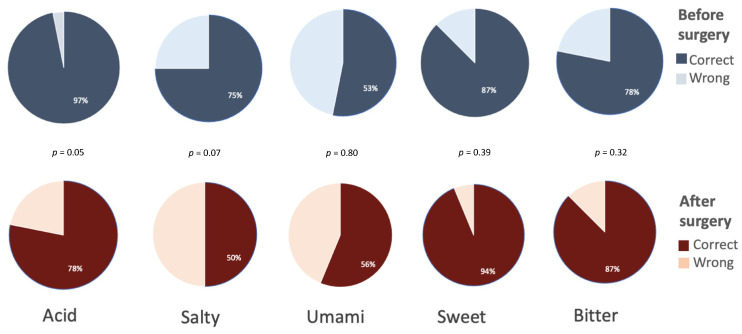
Obese subjects (*n* = 32) correctly identifying the different basic tastes. Data are expressed as percentage of the 32 subjects evaluated at baseline and 1 year after sleeve gastrectomy.

**Figure 4 jcm-11-03957-f004:**
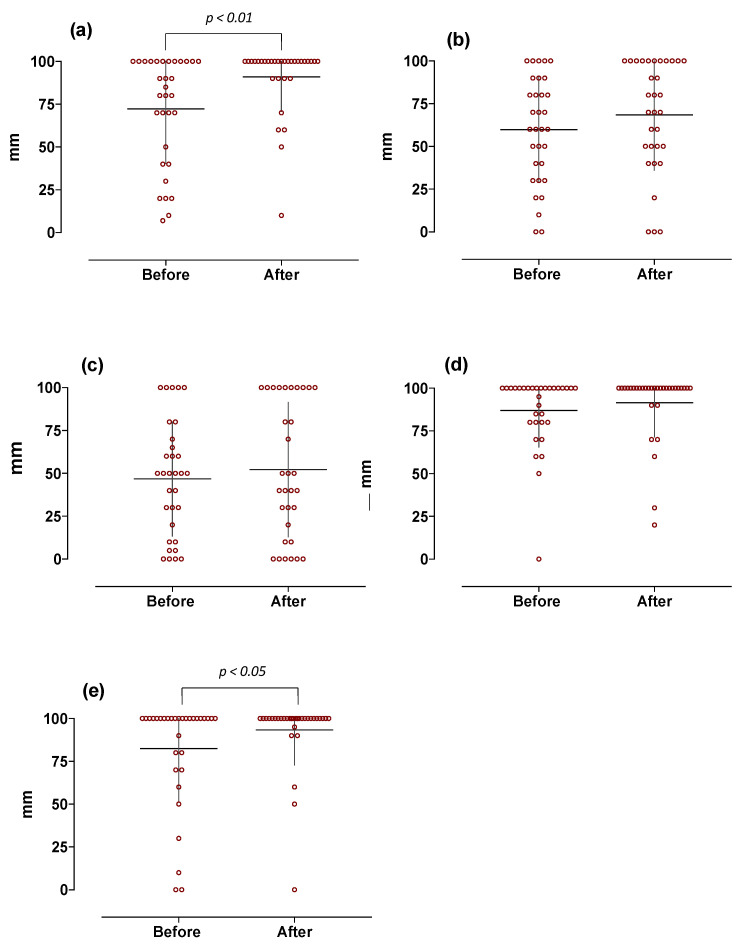
Basic taste-induced intensity in obese subjects at baseline and 1 year after the sleeve gastrectomy (*n* = 32). Graphs are relative to individual scores with superimposed mean ± SD. Data are expressed as mm as measured on 100 mm VAS scale (0 = absent and 100 = maximum perceived intensity). The different panels, respectively, indicate: (**a**) sour-, (**b**) salty-, (**c**) umami-, (**d**) sweet-, and (**e**) bitter-induced responses.

**Figure 5 jcm-11-03957-f005:**
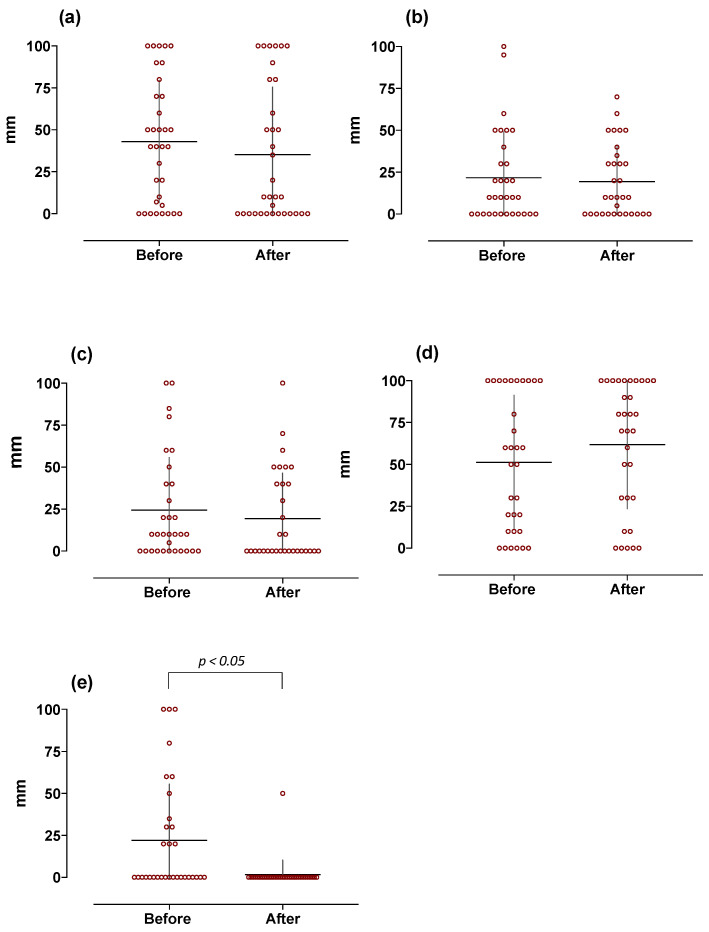
Basic taste-induced pleasantness in obese subjects at baseline and 1 year after sleeve gastrectomy (*n* = 32). Graphs are relative to individual scores with superimposed mean ± SD. Data are expressed as mm as measured on 100 mm VAS scale (0 = unpleasant and 100 = maximum perceived pleasantness). The different panels indicate (**a**) sour-, (**b**) salty-, (**c**) umami-, (**d**) sweet-, and (**e**) bitter-induced responses, respectively.

**Table 1 jcm-11-03957-t001:** Demographics feature of the studied population.

Population (*n*)	Gender (M/F)	Age (Years)	BMI (kg/m^2^)
Controls (40)	14/26	30 ± 11	23 ± 3
Obese (74)	30/44	37 ± 11	46 ± 7.4 *****

* *p* < 0.01 vs. controls; data are expressed as mean ± SD.

**Table 2 jcm-11-03957-t002:** Basic taste-induced intensity in controls and obese subjects. Data are expressed as mean ± SD of scored intensity in mm).

	Sour	Salty	Umami	Sweet	Bitter
Controls (40)	80 ± 16	60 ± 24	50 ± 24	83 ± 24	90 ± 17
Obese (74)	78 ± 28	64 ± 31	51 ± 34	83 ± 22	87 ± 25
*p*	0.65	0.38	0.78	0.87	0.55

## Data Availability

All data generated or analyzed during this study are included in this article. Further inquiries can be directed to the corresponding author.

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
