# Peer review of "Sleeve Gastrectomy-Induced Body Mass Index Reduction Increases the Intensity of Taste Perception’s and Reduces Bitter-Induced Pleasantness in Severe Obesity"

_jcm, 2022, doi:10.3390/jcm11143957_

Round 1

Reviewer 1 Report

In this study, the authors focused on the functional changes of taste identification and taste-induced pleasantness of obese patients who received bariatric surgery. This manuscript records the authors’ observation of the volunteers under stimulation from five basic tastes and may contribute to subsequent studies on taste-derived obesity treatment. However, for the poor quality of figures and data presentation, the authors should substantially revise the manuscript. 

  1. In the ‘Materials and Methods’ section, ‘2.1. Subjects’ part. It is ambiguous when talking about the research subjects. The mean age or BMI seems to be a feature of all males, not all subjects. And I highly suggest a symmetrical writing style, ‘40 healthy’ to ‘74 obese’.
  2. More related data collected from the volunteers would help the readers to assess the real influence of bariatric surgery. Such as an analysis comparing the potential difference between males and females. The information given in this article is not enough.
  3. Line 49, The word ‘morbidly’ should be ‘morbid’ or ‘morbidity’.
  4. Figures should be refined carefully. The p-value of the bitter group should be given in figure 1; The dot and box plot methods, like figures 4 & 5, would be better to show the test results. More details should be given in the figure legends, such as the n number of each group.
  5. In figure 2, the group order of figure 2c should be the same as the others, and the group name should be labeled on each panel.
  6. In figure 3, the acid and salty group seems to be significantly different, please show their p-value or change the pie chart into a dot and box plot. 
  7. There are two ‘figure 3’ in this article.

Author Response

We thank this reviewer for the helpful comment, and we corrected the text according to the suggestion.

  1. More related data collected from the volunteers would help the readers to assess the real influence of bariatric surgery. Such as an analysis comparing the potential difference between males and females. The information given in this article is not enough.

In order to evaluate the gender contribution to the statistical analysis, we run additional analysis and we found no difference between males and females and this is in agreement with what reported by others.

  1. Line 49, The word ‘morbidly’ should be ‘morbid’ or ‘morbidity’.

We corrected.

  1. Figures should be refined carefully. The p-value of the bitter group should be given in figure 1; The dot and box plot methods, like figures 4 & 5, would be better to show the test results. More details should be given in the figure legends, such as the n number of each group.

We thank this reviewer, and we improved the quality of the figures. We also provided more details in the figure legends.

  1. In figure 2, the group order of figure 2c should be the same as the others, and the group name should be labeled on each panel.

We thank the reviewer and according to this suggestion and we redraw the figure 2 as dot and box plot.

  1. In figure 3, the acid and salty group seems to be significantly different, please show their p-value or change the pie chart into a dot and box plot.

We agree with the reviewer’s point. Although a tendency was observed in the respective groups the results were not significantly different. We indicated the exact p values in the revised manuscript.  As the suggestion to use dot and box plot we believe that it is not appropriate to express percentage of subjects.

  1. There are two ‘figure 3’ in this article.

We corrected

Reviewer 2 Report

The Authors attempted to aimed to evaluate basic tastes identification, sensitivity to these tastes and pleasantness of these tastes in obese individuals before and after bariatric surgery. My main objection is that there are many original and review articles on this subject already published, for example:

·         Gero D, Dib F, Ribeiro-Parenti L, Arapis K, Chosidow D, Marmuse JP. Desire for Core Tastes Decreases After Sleeve Gastrectomy: a Single-Center Longitudinal Observational Study with 6-Month Follow-up. Obes Surg. 2017 Nov;27(11):2919-2926. doi: 10.1007/s11695-017-2718-2, 

·         Altun H, Hanci D, Altun H, Batman B, Serin RK, Karip AB, Akyuz U. Improved Gustatory Sensitivity in Morbidly Obese Patients After Laparoscopic Sleeve Gastrectomy. Ann Otol Rhinol Laryngol. 2016 Jul;125(7):536-40. doi: 10.1177/0003489416629162, 

·         Schiavo L, Aliberti SM, Calabrese P, Senatore AM, Severino L, Sarno G, Iannelli A, Pilone V. Changes in Food Choice, Taste, Desire, and Enjoyment 1 Year after Sleeve Gastrectomy: A Prospective Study. Nutrients. 2022 May 14;14(10):2060. doi: 10.3390/nu14102060.

·         Al-Alsheikh AS, Alabdulkader S, Johnson B, Goldstone AP, Miras AD. Effect of Obesity Surgery on Taste. Nutrients. 2022 Feb 18;14(4):866. doi: 10.3390/nu14040866,

·         Shoar S, Naderan M, Shoar N, Modukuru VR, Mahmoodzadeh H. Alteration Pattern of Taste Perception After Bariatric Surgery: a Systematic Review of Four Taste Domains. Obes Surg. 2019 May;29(5):1542-1550. doi: 10.1007/s11695-019-03730-w, 

·         Althumiri NA, Basyouni MH, Al-Qahtani FS, Zamakhshary M, BinDhim NF. Food Taste, Dietary Consumption, and Food Preference Perception of Changes Following Bariatric Surgery in the Saudi Population: A Cross-Sectional Study. Nutrients. 2021 Sep 27;13(10):3401. doi: 10.3390/nu13103401,

to name a few. The majority of them were not cited. For this reason, this study is not novel concerning the majority of tastes and outcomes; however, the relative novelty is the evaluation of the umami taste.

Minor points:

·         Morbid obesity is defined when BMI is 40 kg/m2 or 35 kg/m2 plus one or more co-morbid conditions, but not 35 kg/m2, as stated in the introduction.

·  In the introduction, only taste preferences are mentioned as factors influencing food intake and obesity, while other factors such as hormonal and adipokine imbalance, eating habits, food availability, etc., should be at least mentioned.

·         I want to ensure that the obese subjects examined before and after bariatric surgery are the same, not two different cohorts. This should be clearly stated in the Methods section

·      The relationship between taste/food preferences and obesity is bi-directional (what is first?). Shouldn’t it also be discussed?

Author Response

Reviewer 2

The Authors attempted to aimed to evaluate basic tastes identification, sensitivity to these tastes and pleasantness of these tastes in obese individuals before and after bariatric surgery. My main objection is that there are many original and review articles on this subject already published, for example:

  • Gero D, Dib F, Ribeiro-Parenti L, Arapis K, Chosidow D, Marmuse JP. Desire for Core Tastes Decreases After Sleeve Gastrectomy: a Single-Center Longitudinal Observational Study with 6-Month Follow-up. Obes Surg. 2017 Nov;27(11):2919-2926. doi: 10.1007/s11695-017-2718-2, 
  • Altun H, Hanci D, Altun H, Batman B, Serin RK, Karip AB, Akyuz U. Improved Gustatory Sensitivity in Morbidly Obese Patients After Laparoscopic Sleeve Gastrectomy. Ann Otol Rhinol Laryngol. 2016 Jul;125(7):536-40. doi: 10.1177/0003489416629162, 
  • Schiavo L, Aliberti SM, Calabrese P, Senatore AM, Severino L, Sarno G, Iannelli A, Pilone V. Changes in Food Choice, Taste, Desire, and Enjoyment 1 Year after Sleeve Gastrectomy: A Prospective Study. Nutrients. 2022 May 14;14(10):2060. doi: 10.3390/nu14102060.
  • Al-Alsheikh AS, Alabdulkader S, Johnson B, Goldstone AP, Miras AD. Effect of Obesity Surgery on Taste. Nutrients. 2022 Feb 18;14(4):866. doi: 10.3390/nu14040866, 
  • Shoar S, Naderan M, Shoar N, Modukuru VR, Mahmoodzadeh H. Alteration Pattern of Taste Perception After Bariatric Surgery: a Systematic Review of Four Taste Domains. Obes Surg. 2019 May;29(5):1542-1550. doi: 10.1007/s11695-019-03730-w, 
  • Althumiri NA, Basyouni MH, Al-Qahtani FS, Zamakhshary M, BinDhim NF. Food Taste, Dietary Consumption, and Food Preference Perception of Changes Following Bariatric Surgery in the Saudi Population: A Cross-Sectional Study. Nutrients. 2021 Sep 27;13(10):3401. doi: 10.3390/nu13103401,

to name a few. The majority of them were not cited. For this reason, this study is not novel concerning the majority of tastes and outcomes; however, the relative novelty is the evaluation of the umami taste.

We thank the reviewer for the comment. Although some of the mentioned articles were already present in our list,  we uploaded the lacking references according to the suggestion.

Minor points:

  • Morbid obesity is defined when BMI is ≥ 40 kg/m2or ≥ 35 kg/m2 plus one or more co-morbid conditions, but not ≥ 35 kg/m2, as stated in the introduction.

We corrected.

  • In the introduction, only taste preferences are mentioned as factors influencing food intake and obesity, while other factors such as hormonal and adipokine imbalance, eating habits, food availability, etc., should be at least mentioned.

Again, we have to thank the reviewer for the helpful comment. We better specified this issue in the revised manuscript.

  • I want to ensure that the obese subjects examined before and after bariatric surgery are the same, not two different cohorts. This should be clearly stated in the Methods section

We better specified this point in the methods section of the revised manuscript.   

The relationship between taste/food preferences and obesity is bi-directional (what is first?). Shouldn’t it also be discussed?

This is a very good point. According to our results showing no differences in taste functions between control and obese subjects it seems that changes in basic taste perception is a prerogative of weight-loss.

Unfortunately we are able to answer such intriguing issue, but studies combining fMRI, hormonal in obese likely indicate that an alteration in the food-induced hedonic/adversative  are of major relevance. This issue is discussed in the conclusions of our manuscript.

Round 2

Reviewer 1 Report

1. Please re-size the p values of Figure 1. They are currently too small.

2. Table 2: 0,55 should be 0.55

3. Figure 2: “Controls” and “Obese” sould align with the dots above.

4. Figure 4 and 5: Re-size all the panels. They are currently too small.

Reviewer 2 Report

I am not particularly impressed with how the introduction and discussion were completed. I expected more detailed and extensive information.
